# Mitochondrial Transfer by Human Mesenchymal Stromal Cells Ameliorates Hepatocyte Lipid Load in a Mouse Model of NASH

**DOI:** 10.3390/biomedicines8090350

**Published:** 2020-09-14

**Authors:** Mei-Ju Hsu, Isabel Karkossa, Ingo Schäfer, Madlen Christ, Hagen Kühne, Kristin Schubert, Ulrike E. Rolle-Kampczyk, Stefan Kalkhof, Sandra Nickel, Peter Seibel, Martin von Bergen, Bruno Christ

**Affiliations:** 1Applied Molecular Hepatology Laboratory, Department of Visceral, Transplant, Thoracic and Vascular Surgery, University of Leipzig Medical Center, 04103 Leipzig, Germany; hsumeiju@gmail.com (M.-J.H.); madlen.christ@medizin.uni-leipzig.de (M.C.); hagen.kuehne@freenet.de (H.K.); sandra.brueckner@medizin.uni-leipzig.de (S.N.); 2Department of Molecular Systems Biology, Helmholtz Centre for Environmental Research (UFZ), 04318 Leipzig, Germany; isabel.karkossa@ufz.de (I.K.); kristin.schubert@ufz.de (K.S.); ulrike.rolle-kampczyk@ufz.de (U.E.R.-K.); stefan.kalkhof@hs-coburg.de (S.K.); martin.vonbergen@ufz.de (M.v.B.); 3Molecular Cell Therapy, Center for Biotechnology and Biomedicine, Leipzig University, 04103 Leipzig, Germany; ingo.schaefer@bbz.uni-leipzig.de (I.S.); peter.seibel@bbz.uni-leipzig.de (P.S.); 4Institute for Bioanalysis, University of Applied Sciences Coburg, 96450 Coburg, Germany; 5Department of Therapy Validation, Fraunhofer Institute for Cell Therapy and Immunology, 04103 Leipzig, Germany; 6Institute of Biochemistry, Leipzig University, 04103 Leipzig, Germany

**Keywords:** non-alcoholic steatohepatitis (NASH), tunneling nanotubes (TNTs), primary hepatocytes, organelle transfer, mesenchymal stromal cells

## Abstract

Mesenchymal stromal cell (MSC) transplantation ameliorated hepatic lipid load; tissue inflammation; and fibrosis in rodent animal models of non-alcoholic steatohepatitis (NASH) by as yet largely unknown mechanism(s). In a mouse model of NASH; we transplanted bone marrow-derived MSCs into the livers; which were analyzed one week thereafter. Combined metabolomic and proteomic data were applied to weighted gene correlation network analysis (WGCNA) and subsequent identification of key drivers. Livers were analyzed histologically and biochemically. The mechanisms of MSC action on hepatocyte lipid accumulation were studied in co-cultures of hepatocytes and MSCs by quantitative image analysis and immunocytochemistry. WGCNA and key driver analysis revealed that NASH caused the impairment of central carbon; amino acid; and lipid metabolism associated with mitochondrial and peroxisomal dysfunction; which was reversed by MSC treatment. MSC improved hepatic lipid metabolism and tissue homeostasis. In co-cultures of hepatocytes and MSCs; the decrease of lipid load was associated with the transfer of mitochondria from the MSCs to the hepatocytes via tunneling nanotubes (TNTs). Hence; MSCs may ameliorate lipid load and tissue perturbance by the donation of mitochondria to the hepatocytes. Thereby; they may provide oxidative capacity for lipid breakdown and thus promote recovery from NASH-induced metabolic impairment and tissue injury.

## 1. Introduction

Obesity is a prevalent health problem worldwide, which has been attributed mainly to Western-style diets in combination with reduced physical activity. It is often associated with metabolic co-morbidities, such as diabetes type 2 and non-alcoholic fatty liver diseases (NAFLD), the latter of which has a global prevalence of 24% and is currently the leading cause of chronic liver disease in Europe and in the USA [1]. NAFLD may progress from simple steatosis to chronically inflammatory diseases like non-alcoholic steatohepatitis (NASH), cirrhosis, and hepatocellular carcinoma (HCC) [2,3]. On the cellular level, hepatocyte mitochondrial, peroxisomal and microsomal oxidation of fatty acids, and basal lipophagy [4] are involved in the utilization of hepatic lipids. However, impairment of lipid metabolism may cause an imbalance of utilization and storage, eventually contributing to hepatocyte lipid overload. Lipotoxicity induces endoplasmic reticulum (ER) stress, leading to calcium release from the ER, thus raising cytosolic calcium levels, which in turn interferes with protective autophagy and inhibits the breakdown of lipids, thus aggravating cellular lipid overload [5]. In addition, mitochondrial dysfunction is contributing to the pathogenesis of NAFLD by favoring hepatic lipid storage, and by promoting the production of reactive oxygen species (ROS) as well as lipid peroxidation [6,7,8]. Besides mitochondrial β-oxidation, peroxisomes are involved in long-chain fatty acid β-oxidation and microsomes in ω-oxidation of fatty acids. Like mitochondrial impairment, dysfunction of peroxisomal and microsomal fatty acid metabolism contributes to the pathogenesis of NASH by favoring lipid storage and production of ROS, respectively [9,10].

Pharmacological therapy may address NASH-associated co-morbidities like diabetes [11], yet, liver transplantation in NASH patients has been shown to reduce the risk of progression into HCC [12,13]. Indeed, NASH features the second most common indication for liver transplantation in the USA [1,14]. However, the invasiveness of organ transplantation, lifelong immune suppression, the shortage of donor livers, and eventually the high costs altogether fostered the search for alternative therapy approaches [15,16]. Mesenchymal stromal cells (MSCs) have been demonstrated to ameliorate NASH in rodent animal models [17,18,19]. Transplantation of hepatocyte-like cells derived from human bone marrow mesenchymal stromal cells (MSCs) alleviated lipid load, ameliorated hepatic inflammation as well as fibrosis, and enhanced proliferation of host hepatocytes in an experimental model of NASH in the immunodeficient mouse [20]. In line, transplantation of MSCs decreased fibrosis and activation of hepatic stellate cells in mouse and rat liver cirrhosis triggered by carbon tetrachloride (CCl_4_) [21,22]. MSC restored ammonia and purine metabolism in a mouse model of acute liver failure [23], improved acute liver injury caused by acetaminophen [24], and enhanced liver regeneration in mice and rats after extended hepatectomy [25,26]. Clinically, the application of MSC ameliorated inborn errors of liver metabolism, such as ornithine carbamoyltransferase deficiency [27], decreased the severe bleeding complications in a hemophilia A patient [28], and improved liver function in cirrhotic patients [29,30,31]. However, some studies revealed only transient or even negligible effects of MSC treatment of liver diseases [32,33]. Thus, the identification of the mechanisms involved in MSC action is crucial to improve the therapeutic potential of MSCs.

Here, we showed in a mouse model of NASH and in co-cultures of fat-laden hepatocytes and MSCs that MSCs shifted pathological lipid storage to utilization likely by the transfer of MSC-derived functional mitochondria to the hepatocytes.

## 2. Experimental Section

### 2.1. Experimental Design

Xenotransplantation of human bone marrow-derived mesenchymal stromal cells was performed in immunodeficient Pfp/Rag2^−/−^ mice (C57BL6, B6.129S6-Rag2(tm1Fwa)Prf1(tm1Clrk)N12. Taconic; Ejby, Denmark). At the age of 12 weeks, male mice were either fed a NASH−inducing methionine-choline-deficient diet (MP Biomedicals, Eschwege, Germany), or kept on a standard rodent chow. After five weeks of feeding, all mice underwent 1/3 partial hepatectomy and received 1.5 × 10^6^ human MSCs (differentiated into the hepatocytic lineage) via splenic injection as described before [34]. PBS served as the vehicle control. The four resulting groups represented +NASH+MSC, +NASH-MSC, -NASH+MSC, and -NASH-MSC. The procedures for the isolation and hepatocytic differentiation of MSCs from human bone marrow have been described in detail previously [35], and were approved by the Institutional Ethics Review Board Leipzig (Ethik-Kommission an der Medizinischen Fakultät der Universität Leipzig; file no. 282/11-lk; 1 December 2016). After surgery, respective feeding regimes were continued for another week until mice were sacrificed, and livers harvested. Mice were housed under controlled conditions at a 12-h light/dark cycle, at an ambient temperature of 22 ± 2 °C, and humidity of 50%–60%. All experimental procedures involving animals were approved by the federal authorities of Saxony (Saxon State Directorate Chemnitz; file no. TVV_54_16; 2 May 2017) and followed the guidelines of the animal welfare act.

### 2.2. Proteome and Metabolome Analyses

#### 2.2.1. Liver Metabolite and Protein Extraction and Preparation for Protein Quantification by Stable Isotope Labeling of Mammals (SILAM)

Frozen liver tissue samples (100–200 mg) were homogenized with a ball mill in 500 μL of lysis buffer A (40 mM Tris base, 7 M urea, 4% CHAPS, 100 mM dithiothreitol (DTT), 0.5% (*v*/*v*) biolyte). Benzonase was added and the homogenates were incubated at room temperature for 20 min and afterwards centrifuged at 12.000× *g* for 20 min at 30 °C. The resulting supernatant was collected, and the pellet was again extracted using 500 μL of lysis buffer B (40 mM Tris base, 5 M urea, 2 M thiourea, 4% CHAPS, 100 mM DTT, 0.5% (*v*/*v*) biolyte). After centrifugation, the supernatants of the first and the second extraction were combined. Then, 50 µL of the combined supernatants were desalted to 50 mM ammonium bicarbonate by centrifugal filtration using filtration units (molecular weight cutoff of 10 kDa, Vivacon 500, Sartorius Group, Göttingen, Germany). The permeate was applied for metabolomics analyses. The protein concentrations in the retentate were determined using a detergent-compatible colorimetric protein assay (660 nm Protein Assay, Thermo Scientific–Pierce, Dreieich, Germany). Analogously, a protein extract of a liver of a ^13^C_6_-lysine-labeled black6 mouse (^13^C_6_-lysine, 97%, Cambridge Isotope Laboratories, Inc., Tewksbury, MA, USA) was prepared as an isotope-labeled standard.

#### 2.2.2. In-Gel Tryptic Digestion and Liquid Chromatography-Tandem Mass Spectrometry (GeLC-MS/MS)

Protein mixtures were mixed with 0.5 M Tris-HCl buffer (pH 6.8) containing 40% (*v*/*v*) SDS, 20% (*v*/*v*) glycerol, 2% (*v*/*v*) bromophenol-blue, and 10% (*v*/*v*) 2-mercaptoethanol. After heating for 5 min at 95 °C, a 1-D-SDS-PAGE was carried out using Laemmli’s protocol. Proteins were stained with Coomassie Brillant Blue G-250 and each lane was cut into 10 pieces of equal size. Gel pieces were washed twice in 200 µL of ACN/50 mM ABC (ammonium bicarbonate, Fluka (Thermo Fisher Scientific, Dreieich, Germany), 1:1, (*v*/*v*)) for 10 min at room temperature (RT). Reduction and alkylation of disulfide bonds were performed by adding 10 mM dithiothreitol/10 mM ABC (Thermo Fisher Scientific, Dreieich, Germany) for 30 min at RT followed by incubation in 100 mM iodacetamide/10 mM ABC (Sigma-Aldrich, Taufkirchen, Germany) for 30 min at RT in the dark. After another washing step in 10 mM ABC for 10 min at RT, proteins were digested by incubating each gel slice with 300 ng of trypsin (Promega, Mannheim, Germany) in 50 mM ABC at 37 °C overnight. Proteolysis was quenched by 10% formic acid and proteolytic peptides were extracted. Samples were dried by vacuum centrifugation and resuspended in 20 µL of 0.1% formic acid. Extracted peptides were analyzed by online reversed-phase nanoscale liquid chromatography tandem mass spectrometry on a NanoAcquity UPLC system (Waters Corporation, Milford, MA, USA) connected to an LTQ-Orbitrap XL ETD (Thermo Fisher Scientific, Waltham, MA, USA) and equipped with a chip-based nano ESI source (TriVersaNanoMate, Advion, Ithaca, NY, USA) as described previously [36]. Protein identification and relative quantification were performed using Proteome Discoverer (version 1.4, Thermo Scientific, Bremen, Germany). Oxidation (methionine) and acetylation (protein N-termini) were used as variable modifications, while carbamidomethylation (cysteine) was set as a fixed modification. A database search was carried out by the search engine MASCOT against the UniProt mouse reference proteome (www.uniprot.org, 05-2014). Relative protein quantification was performed based on the measured ratios (treated mouse vs. isotopically labeled mouse standard) of all lysine-containing peptides. Fold changes (FCs) between different groups were calculated in reference to the internal control.

#### 2.2.3. Metabolome

Targeted metabolomics was conducted using the AbsoluteIDQ p150 kit (BIOCRATES Life Sciences AG, Innsbruck, Austria) as described before [26]. In brief, metabolites were extracted from the livers as descripted above, and prepared according to the manufacturer’s protocol [37]. Multiple reaction monitoring was carried out on an Agilent 1100 series binary HPLC system (Agilent Technologies, Waldbronn, Germany) coupled to a 4000 QTRAP (AB Sciex, Darmstadt, Germany) via a TurboIon spray source. The data evaluation for the quantification of metabolite concentrations was performed with the MetIQ software package.

### 2.3. Combined Analysis and Visualization of Proteome and Metabolome Data

#### 2.3.1. Weighted Gene Correlation Network Analysis (WGCNA)

FCs of proteins and metabolites were analyzed in R-3.5.0 with the use of several packages [38,39,40,41,42,43,44,45,46]. Hierarchical clustering was conducted with Euclidean distance measure. To unravel changes for single proteins and metabolites, the data were filtered for analytes that were quantified at least in duplicate for the particular comparison, and the Student’s *t*-test was performed for analytes that were identified at least in triplicate (Appendix A, FCs and *p*-values proteomics, FCs and *p*-values metabolomics). For WGCNA, the average log2-transformed FCs of proteins and metabolites that were quantified at least in duplicate over all data sets (406 proteins and 148 metabolites) were scaled to integer values between 0 and 100 (Appendix A, WGCNA data). The networks were constructed across all the measured samples with the R package WGCNA [47] as described before [48]. The used trait matrix, containing the different comparisons, may be found in the Appendix A, WGCNA trait matrix. The soft power threshold for WGCNA was set to 18 to arrive at the network adjacency. The Topology Overlap Matrix (TOM) was created using a cut height of 0.15 and a minimum module size of 25. The analysis identified 11 modules of co-expressed analytes, identified with different colors. A summary of the analytes that were assigned to each module can be found in the Appendix A, WGCNA module contents. Finally, for each of the obtained modules, significantly enriched KEGG (Kyoto Encyclopedia of Genes and Genomes) pathways were determined using R-3.5.0 [49,50,51] without defining a *p*-value threshold. For this purpose, the mouse database was used as background [52]. Lists of all enriched pathways for each module can be found in the Appendix A, WGCNA KEGG results.

#### 2.3.2. Identification of Key Drivers

Identification of trait-specific key drivers was performed based on the WGCNA results as described before [48]. Therefore, module- and trait-specific gene significances and module memberships were calculated for each analyte. Key drivers were assumed to be analytes with absolute gene significance ≥0.75 and absolute module membership ≥0.75 (Appendix A, WGCNA key drivers). Gene names (Appendix A, FCs and *p*-values proteomics) and KEGG pathways (Appendix A, KEGG pathway mapping) were assigned to proteins using the DAVID Bioinformatics Resources 6.8 [53]. Thus, key drivers for the observed effects were identified. From the whole proteome, proteins that are related to mmu03320: PPAR signaling pathway, mmu04975: Fat digestion and absorption, mmu04610: Complement and coagulation cascades, mmu00190: Oxidative phosphorylation, mmu00480: Glutathione metabolism, mmu04932: Non-alcoholic fatty liver disease (NAFLD), mmu04146: Peroxisome, mmu00061: Fatty acid biosynthesis, mmu00062: Fatty acid elongation, mmu01212: Fatty acid metabolism, and mmu00071: Fatty acid degradation were extracted for deeper insights into effects (Appendix A).

### 2.4. Immunohistochemistry

A comprehensive list of antibodies used throughout this study is given in Appendix A. Mouse livers were fixed in 4% paraformaldehyde overnight and embedded in paraffin. Slices of 1 µm were dewaxed and rehydrated by standard procedures. Heat-mediated epitope retrieval was done in either citrate buffer (pH 6.0) or Tris-EDTA buffer (pH 9.0) for 30 min. Endogenous peroxidases were blocked with 3% hydrogen peroxide/methanol for 20 min followed by a 60-min blocking step in 5% BSA/0.5% Tween20, and additionally a 15-min avidin and 15-min biotin block (SP-2001, Avidin-Biotin Blocking Kit, Vectorlabs, Eching, Germany). Primary antibodies against Cyp2e1 (1:200, ab28146, abcam, Cambridge, UK) and 4-HNE (1:500, HNE11-S, Alpha Diagnostic International, San Antonio, TX, USA) were applied overnight at 4 °C, subsequently coupled to biotin-labelled secondary goat anti-rabbit antibody (1:200, 111-065-003, Dianova, Hamburg, Germany), and visualized by the Vectastain Elite ABC Kit (PK-6100, Vectorlabs, Eching, Germany) followed by DAB chromogen (Thermo Fisher Scientific, Dreieich, Germany) incubation. Hematoxylin was used as a nuclei counterstain. For immunofluorescence, slices were blocked in 5% goat serum/PBS (ccpro, Oberdorla, Germany) for 20 min and in 5% BSA/0.5% Tween20 for 60 min. Sections were subsequently incubated with primary antibodies against CD36 (1:100, NB400-144, Novusbio, Wiesbaden-Nordenstadt, Germany), E-cadherin (1:200, BD 610182, eBioscience, Heidelberg, Germany), and β-catenin (1:500, BD 610154, eBioscience, Heidelberg, Germany) overnight at 4 °C. Corresponding secondary antibodies, i.e., goat anti-rabbit AlexaFlour 568 (1:200, A11036, Life Technologies, Ober-Olm, Germany) to CD36, goat-anti-mouse Cy3 (1: 300, 115-165-003, Dianova, Hamburg, Germany) to E-cadherin, and goat anti-mouse AlexaFlour 488 (1:300, 115-545-003, Dianova, Hamburg, Germany) to β-catenin, were applied for 1 h at room temperature followed by nuclear staining with DAPI (1:1000, Carl Roth GmbH + Co. KG, Karlsruhe, Germany) for 5 min, respectively. Slides were mounted with 50% glycerol solution and lacquer, and images taken using the Zeiss Axio Observer.Z1 microscope. For the co-stain of human mitochondria in mouse hepatocytes, antigen retrieval using dewaxed slices was performed by heating for 30 min in citrate buffer (14746, SignalStain unmasking solution, Cell Signaling Technology, Frankfurt/Main, Germany) and subsequent cooling on ice for 30 min. After two washings in PBS, slices were blocked for 80 min using 5% goat serum and 0.3% TritonX-100 in PBS. Primary antibodies against mouse cyclophilin A (1:100, 2175, Cell Signaling Technology, Frankfurt/Main, Germany) and anti-human mitochondria (1:200, MAB 1273, Millipore, Darmstadt, Germany) were added overnight at 4 °C, and slices washed 3 times with PBS thereafter. The first secondary antibody (1:200 goat anti-mouse Cy3; 115-165-003, Dianova, Hamburg, Germany) was added for 60 min at room temperature and slices were washed 2 times for 10 min each with PBS. The second secondary antibody (1:500 goat anti-rabbit AlexaFluor 488, 4412, Cell Signaling Technology, Frankfurt/Main, Germany) was added for 60 min at room temperature. After 3 washings for 5 min each with PBS, nuclei were stained with DAPI and slides finally mounted with 50% glycerol solution and lacquer. Images were taken using the Zeiss Axio Observer.Z1 microscope equipped with ApoTome.2 at 40× magnification.

### 2.5. Western Blotting

In total, 30–40 mg of liver tissue were lysed in RIPA buffer (50 mM Tris, 150 mM NaCl, 0.1% SDS, 1% Triton X-100, 1 mM EDTA+EGTA, 0.5% Na-deoxycholate, pH 7.5) supplemented with protease inhibitors (Roche, Mannheim, Germany). Crude lysates were centrifuged at 13,000 rpm for 15 min and the clear supernatant was collected. The protein concentration was determined by the bicinchoninic acid assay. Then, 50 μg of protein were subjected to standard SDS gel electrophoresis and blotted onto PVDF membranes. Non-specific binding was blocked with 5% skim milk (vinculin), or 5% BSA (PPARα, CD36) in Tris-buffered saline Tween-20 (TBS-T) for two hours. The primary and secondary antibodies used were as follows: anti-PPARα (1:1000, MAI-822 Thermo Fischer Scientific, Dreieich, Germany), anti-CD36 (1:1000, NB400-144, Novusbio, Wiesbaden-Nordenstadt, Germany), anti-Vinculin (1:3000, 05-386, Merck, Darmstadt, Germany), anti-rabbit-HRP (1:7500, BD 554021, eBioscience, Heidelberg, Germany), and anti-mouse-HRP (1:7500, BD 554002, eBioscience, Heidelberg, Germany). Blots were developed using the enhanced chemiluminescence Prime reagent kit (GE Healthcare, Buckinghamshire, UK). Vinculin was used for the normalization and calculation of the relative abundances.

### 2.6. Quantification of Liver Triglyceride Content

In total, 100 mg of liver tissue were homogenized in 1 mL of 5% Igepal (I3021, Sigma-Aldrich, Taufkirchen, Germany) using a pestle and mortar. Samples were heated for 4 min at 95 °C in a thermomixer, cooled at room temperature, heated again, and then centrifuged using a microcentrifuge at maximal speed. Supernatants were collected and diluted 1:10 with distilled water. Then, 25 µL of the samples were pre-warmed at 37 °C for 1–5 min and subjected to the Triglyceride Assay Kit-Quantification (ab65336, Abcam, Berlin, Germany), essentially as described by the supplier.

### 2.7. Isolation of Primary Mouse Hepatocytes and Co-Culture with Hepatocytic Differentiated MSCs

Primary hepatocytes (HCs) were isolated from male Pfp/Rag2^−/−^ mouse livers by the two-step liver perfusion with collagenase (NB4G, Serva Electrophoresis GmbH, Heidelberg, Germany) as described [54]. HCs and hepatocytic differentiated MSCs at the cell number ratios as indicated were initially seeded onto collagen-coated dishes at a total cell density of 40,000 cells/cm^2^ and grown for 3 h in minimal essential medium (MEM) (Merck, Darmstadt, Germany) containing 2% fetal calf serum (FCS) (Gibco, Darmstadt, Germany) to allow for attachment. The medium was replaced by either standard hepatocyte growth medium (HGM, [55]) supplemented with EGF and HGF (20 ng/mL each) serving as the control, or two different steatosis-inducing media: methionine-choline-deficient (MCD, c.c.pro Oberdorla, Germany) or HGM supplemented with 0.5 mM palmitic acid (C16:0). To identify potential paracrine effects mediated by the MSCs, conditioned media were collected from MSCs, hepatocytes, and co-cultures of both after 2 days of culture, centrifuged at 270× *g* to remove cell debris, and subsequently used to treat hepatocytes under control (HGM) and treatment conditions (MCD or C16:0) for an additional 1 and 2 days. Phase contrast pictures were captured with a Primovert inverted microscope with the Zen software (Zeiss, Jena, Germany).

### 2.8. Cell Labeling with Fluorescent Vital Dyes

To highlight MSCs in co-culture with hepatocytes using a fluorescence microscope, MSCs in suspension were pre-labeled prior to seeding with either 5-(and 6)-carboxyfluorescein diacetate succinimidyl ester (CFSE; Ex/Em: 494/521) or CellTrace Yellow (Ex/Em: 546/579) at a final concentration of 5 μM by shaking in the dark at 37 °C for 30 min. Cells were subsequently washed with PBS containing 20% of FCS, followed by two washes with 5% FCS [56]. Fluorescent dyes as mentioned above were from Thermo Fisher Scientific, Dreieich, Germany. To label mitochondria, MSCs in suspension were stained with 500 nM of MitoTracker Deep Red (Ex/Em: 644/665) or 150 nM of MitoTracker Red CMXRos (Ex/Em: 579/599) in PBS at 37 °C for 45 min with gentle shaking, followed by 3 washes with PBS. MitoTracker dyes were kindly provided by Prof. Dr. Lea Ann Dailey and Dr. Lysann Tietze, Institute of Pharmacy, University of Halle-Wittenberg, Halle, Germany. Labeled cells in suspension were kept in the dark on ice prior to further use.

### 2.9. Fluorescence Staining after Cell Fixation

Cells grown on coverslips were fixed with 3.7% formaldehyde at room temperature for 15 min, followed by 3 washings with PBS. Neutral triglycerides and lipids were labeled with Oil red O (ORO) as previously described [57]. For immunostaining, fixed cells were permeabilized with 0.1% Triton X-100 for 5 min at RT, followed by blocking with PBS containing 5% BSA for 1 h and additional blocking with 5% normal goat serum for 1 h. Cells were further stained with the mouse anti-human mitochondria antibody (1:400, MAB1273, Millipore, Darmstadt, Germany), or the rabbit anti-apoptosis-inducing factor (AIF) antibody (1:400, Rabbit mAb 5318, Cell Signaling Technology, Frankfurt/Main, Germany, kindly provided by Prof. Dr. Gabriela Aust, Department of Visceral, Transplant, Thoracic and Vascular Surgery, University of Leipzig Medical Center, Leipzig, Germany) in 1% BSA overnight at 4 °C. The following day, cells were incubated with goat anti-mouse antibodies conjugated with Cy3 (115-165-003, Dianova, Hamburg, Germany) or goat anti-rabbit antibodies conjugated with Alexa Fluor 488 (A11008, Thermo Fisher Scientific, Dreieich, Germany). To visualize cell morphology and discriminate between MSCs and hepatocytes, cells were stained with 0.1% CytoPainter Phalloidin-iFlour405 (ab176752, Abcam, Berlin, Germany) in 1% BSA in PBS for 1.5 h at RT to label F-actin. Where indicated, nuclei were counterstained with 1 μg/mL of 4,6-diamidino-2-phenylindole (DAPI).

### 2.10. Image Capture and Analysis

Images were taken using the Zeiss Axio Observer.Z1 inverted microscope at 20× magnification, or with ApoTome.2 at 40×. Lipid content was quantified by image analysis using the ImageJ software (ImageJ 1.42, National Institutes of Health, Bethesda, MD, USA). Results were normalized as the percentage amount of stain/100 HCs out of 6-10 microscopic visual fields per group. Results from 4 independent experiments are expressed as mean ± standard deviation (SD). Pictures of time-lapse live cell imaging were captured at an interval of 15 min using a laser scanning confocal microscope (Leica Microsystems, Wetzlar, Germany) at a magnification of 20×. One-μm-thick sections were acquired with a total z volume of 25 μm, and the maximum intensity projections are presented (cf. Figure 5B,C, and Appendix A).

### 2.11. Morphological Subtyping of Mitochondria

The cells were stained with 150 nM of MitoTracker Red CMXRos as described under ‘Cell labeling with fluorescent vital dyes’, followed by fixation. Pictures were captured using the Zeiss Axio Observer.Z1 inverted microscope equipped with ApoTome.2 with a 40× objective. Intact cells were cropped using Adobe Photoshop CS6 (v. 13.0, München, Germany). On average, the mitochondrial morphology of 18.31 hepatocytes and 12.58 MSC per group was analyzed by the automatic subtyping and quantification software MicroP, as developed and described by Peng et al. [58].

### 2.12. RNA Isolation, cDNA Synthesis and PCR

RNA was isolated using the QIAzol Lysis Reagent (Qiagen, Hilden, Germany). The cDNA was synthesized using the Maxima H Minus First Strand cDNA Synthesis Kit (Thermo Fischer Scientific, Dreieich, Germany). PCR was carried out using the PCR Master Mix (2×) (Thermo Fischer Scientific, Dreieich, Germany) and appropriate primer pairs as listed in Appendix A. PCR products were separated in Tris/borate/EDTA (TBE) agarose gels and quantified with ImageJ. Beta-2-microglobulin (B2M) was used for normalization.

### 2.13. Statistical Analysis

Results are shown as means ± standard deviation (SD) from 3–7 livers in each group run in analytical duplicates unless otherwise indicated. Statistical comparisons, if not otherwise indicated, were made using either the paired *t*-test or two-way ANOVA. Differences between groups were considered significant at a *p* value of ≤0.05.

## 3. Results

We and others have shown that the transplantation of mesenchymal stromal cells into mice suffering from NASH attenuated the lipid load, reduced inflammation, and resolved fibrosis [18,34,59,60]. Yet, the mechanism behind remained mainly elusive. Here, we aimed by omics approaches in combination with network analyses to identify pathways and key players in NASH, which were affected by MSC treatment. Potential mechanisms of MSC action as deduced from the WGNCA were investigated in co-cultures of hepatocytes and MSCs.

### 3.1. WGCNA Suggests a Shift from Lipid Storage to Utilization by MSCs

Based on the protein and metabolite intensities, which were obtained by applying untargeted proteomics and targeted metabolomics to mouse livers, ratios were calculated for the following comparisons to get insights into the mechanisms of the MSC treatment: +NASH-MSC vs. -NASH-MSC, +NASH+MSC vs. -NASH-MSC, -NASH+MSC vs. -NASH-MSC, +NASH+MSC vs. +NASH-MSC, +NASH-MSC vs. -NASH+MSC, and +NASH+MSC vs. -NASH+MSC. Those values were used for further analyses.

By WGCNA, co-expressed proteins and metabolites were summarized into modules, followed by correlation of the obtained module eigengenes (modules first principal component) with traits. In total, 11 modules were identified and the results of the correlation with selected traits are shown in Figure 1A, while the complete module–trait correlation can be found in Appendix A. For each of the obtained modules, significantly enriched pathways were determined using KEGG (Kyoto Encyclopedia of Genes and Genomes, Appendix A, WGCNA KEGG results). Anti-correlations for NASH and MSCs are observable for the magenta, turquoise, black, and blue module, showing significant (*p*-value ≤ 0.05) enrichment of KEGG pathways related to amino acid biosynthesis and central carbon metabolism (Appendix A). Additionally, in the case of +NASH-MSC vs. -NASH-MSC and +NASH+MSC vs. +NASH-MSC, the described anti-correlation was observable but for the red, brown, green, green-yellow, yellow, and purple module that also showed significant enrichment of pathways connected to amino acid metabolism and central carbon metabolism, and in addition also to fatty acid degradation and metabolism as well as peroxisome proliferator-activated receptor (PPAR) signaling. The PPAR signaling showed the highest enrichment in the brown module, which positively correlated with +NASH-MSC vs. -NSH-MSC and negatively with +NASH+MSC vs. +NASH-MSC. This indicates that proteins that are assigned to this pathway in KEGG tended to show increased abundances upon NASH and decreased abundances upon MSC treatment. Interestingly, the comparison of +NASH+MSC vs. -NASH-MSC did not lead to high correlations, neither in the brown module containing PPAR signaling pathway-related proteins, nor in most of the other modules, indicating that the MSCs were successfully used to treat NASH, resulting in expression profiles similar to what was observable in the control group (Figure 1A).

Furthermore, a key driver analysis was conducted for the traits NASH and +NASH+MSC vs. +NASH-MSC based on the WGCNA results (Appendix A, WGCNA key drivers). This analysis allowed the identification of analytes that were highly connected to the particular modules and traits, suggesting their critical role as mediators for the observed effects (Appendix A). KEGG pathways were assigned to all, within this study, identified proteins (Appendix A, KEGG pathway mapping) to identify the functions of the selected key drivers. Thereby, we focused on the following KEGG pathways: Non-alcoholic fatty liver disease (NAFLD), PPAR signaling pathway, fat digestion and absorption, complement and coagulation cascades, oxidative phosphorylation, glutathione metabolism, peroxisome, fatty acid biosynthesis, fatty acid elongation, fatty acid metabolism, and fatty acid degradation (Appendix A). Figure 1B shows Log2(FCs) and *p*-values for a selection of the obtained key drivers. Apparently, metabolites gave information about NASH effects and more importantly also about effects upon MSC treatment. This was also true for the identified key driver proteins that furthermore gave insights into affected pathways. Proteins that were identified to be key drivers were mainly related to cellular stress (e.g., Sod2, Gstm2, Mgst1, Gstp1), as indicated by their contribution to the KEGG pathways glutathione metabolism and peroxisome. Furthermore, lipid metabolism was affected (e.g., Apob, Fasn), with an assignment of key drivers to fat digestion and absorption as well as fatty acid biosynthesis and metabolism. Interestingly, several of the identified key drivers showed opposite Log2(FCs) for +NASH+MSC vs. +NASH-MSC compared to the other investigated ratios, indicating that the MSC treatment of NASH compensated the NASH effects. In summary, the WGCNA showed an anti-correlation of NASH effects and effects upon MSC treatment as well as only minor effects for the MSC-treated NASH as compared to the control group. This indicated that MSCs reversed NASH effects, which was also confirmed by the expression profiles of the identified key drivers.

The hepatic acute-phase reaction is the defense response of the liver to injury, trauma, or inflammation [61]. It is associated with an increase of the expression of ‘positive’ acute-phase proteins (APP) on the expense of the ‘negative’ APPs [61]. The proteome analysis showed that the APPs Serpina1c (alpha-1-antitrypsin), which can be assigned to complement and coagulation cascades in KEGG, and several Mup (major urinary proteins) were significantly increased in +NASH-MSC livers as compared to control livers (Figure 1C). This increase was ameliorated in NASH livers receiving MSCs to levels comparable to the control group. Moreover, analysis of differentially expressed proteins revealed that compared to healthy (-NASH-MSC) control livers, +NASH-MSC livers showed a marked decrease in the expression of mitochondrial proteins Ndufa7 (NADH dehydrogenase 1 alpha subunit 7, KEGG: Oxidative phosphorylation, NAFLD), Cox2 (cytochrome c oxidase subunit 2, KEGG: Oxidative phosphorylation, NAFLD), and Hadh (hydroxyacyl-CoA dehydrogenase, KEGG: Fatty acid elongation, degradation, and metabolism). Ndufa7 and Cox2, as parts of complexes I and IV of the respiratory chain, and Hadh, as a key factor in the metabolism of short-chain fatty acids, are directly involved in mitochondrial β-oxidation and energy production. In addition, cytosolic proteins Acsl5 (long-chain fatty acid-CoA (Coenzyme A) ligase 5, KEGG: Fatty acid biosynthesis, degradation, and metabolism, PPAR signaling pathway, peroxisome) and Fabp1 (liver-specific fatty acid-binding protein 1, KEGG: PPAR signaling pathway, fat digestion and absorption) were increased in +NASH-MSC livers, indicating the stimulation of fatty acid activation towards triglyceride synthesis. Furthermore, the peroxisomal Acaa1a (acetyl-coenzyme A acyltransferase 1, KEGG: Fatty acid degradation and metabolism, PPAR signaling pathway) showed decreased abundances in NASH livers, which might be associated with the reduction of peroxisomal lipid oxidation. In addition, the fat digestion and absorption-related proteins Apoa4 (apolipoprotein A4) and Apob (apolipoprotein B100) showed the opposite behavior in +NASH-MSC livers compared to MSC-treated NASH livers, with Apoa4 being decreased in NASH livers compared to controls, whilst Apob was increased. This indicates an impairment of lipoprotein secretion (Figure 1C). Taken together, in the NASH livers, lipid metabolism seemed to be shifted from utilization to storage. As a hallmark of NASH, liver lipid deposition is increased due to the excess provision of fatty acids from the adipose tissue in conjunction with the impairment of hepatocyte mitochondrial β-oxidation, involving a decrease in respiratory chain activity [62], and impairment of the secretion of triglycerides via very low density lipoproteins (VLDL). The latter has been shown to be hampered due to decreased expression of the microsomal triglyceride transfer protein (MTTP), a chaperone involved in VLDL assembly [63]. The proteomic changes as described above and the decrease in serum and increase in liver triglycerides, respectively, as shown in our previous study in this NASH model [34], may suggest that the impairment of fatty acid oxidation and triglyceride secretion contributed, at least in part, to hepatosteatosis and inflammation in livers of the MCD diet-fed mice and amelioration by MSC treatment. Since mitochondrial impairment might be a major cause of lipid accumulation, we anticipated that MSC treatment might improve mitochondrial function.

Since MSCs, besides others, reversed the NASH-induced decrease of Acaa1a, and PPAR signaling was among the significantly enriched pathways in the modules that showed an anti-correlation between +NASH-MSC vs. -NASH-MSC and +NASH+MSC vs. +NASH-MSC, we speculated that this pathway could play a role in the attenuation of pathological lipid storage in NASH livers. Thus, we analyzed the expression of the peroxisome proliferator-activated receptor alpha (PPARα), the key regulator of peroxisomes and liver lipid metabolism [64]. Semi-quantitative Western blot analysis revealed that PPARα was suppressed to low levels in +NASH-MSC livers. Upon MSC transplantation, PPARα was re-expressed, albeit to a significantly lower extent than in controls (-NASH-MSC), corroborating that peroxisomal lipid metabolism may potentially contribute to MSC-mediated lipid clearance (Figure 2A). In addition to the uptake by Fatp1, fatty acids are transported into hepatocytes by the fatty acid translocase CD36. Semi-quantitative Western blot analysis of CD36 revealed that expression was upregulated in +NASH-MSC as compared to -NASH-MSC livers, which was not impacted by MSC treatment (Figure 2A). As verified by fluorescence immunohistochemistry, it seemed that predominant peripheral, presumably membranous localization of CD36 in +NASH-MSC livers changed to predominant cytoplasmic localization in +NASH+MSC-livers (Figure 2B). Since translocase activity of CD36 is associated with membrane localization [65], the cytoplasmic shift by MSC treatment might indicate the attenuation of fatty acid uptake.

In summary, we reason that MSC alleviated the lipid burden in NASH livers by the improvement of lipid utilization by mitochondria in conjunction with the mutual balance of fatty acid import and lipid export.

### 3.2. Improvement of Tissue Homeostasis by MSCs

NASH is associated with an increased expression of Cyp2e1, a cytochrome P450 family member involved in the metabolism of polyunsaturated fatty acids [66]. The enzyme produces reactive oxygen species, leading to lipid peroxidation and formation of the byproduct 4-hydroxynonenal (4-HNE) [67]. To evaluate, whether oxidative stress was increased in NASH livers, we analyzed 4-hydroxynonenal (4-HNE) and Cyp2e1 by immunohistochemistry. In control livers, Cyp2e1 was zonally expressed in pericentral hepatocytes both untreated (-NASH-MSC) and treated with MSCs (-NASH+MSC), which was paralleled by weak zonal detection of 4-HNE. In NASH livers, both Cyp2e1 and 4-HNE were increased particularly in perivenous regions displaying a high fat content. Treatment with MSCs lowered both again to levels comparable to controls (Figure 3A), suggesting that MSCs ameliorated oxidative stress associated with enhanced lipid oxidation in NASH livers.

Liver function is strongly dependent on the polar orientation of hepatocytes along the sinusoids, i.e., facing the bile canaliculi with their apical and the sinusoids with their basolateral side. Chronic liver damage induces hepatocyte epithelial-mesenchymal transitions (EMTs) with a loss of hepatocyte polarity and eventually functionality [68]. The periportal expression of E-cadherin (E-cad), indicative for the epithelial organization of the hepatic parenchyma, was abrogated in NASH mouse livers as shown before [69]. Here, we confirmed this finding by immunohistochemical detection of E-cad and β-catenin (β-cat) co-localizing in the cell membrane at adherens junctions. In -NASH-MSC livers, both proteins were co-localized in periportal hepatocytes, indicating intact adherens junctions and hepatocyte polarity. In +NASH-MSC livers, however, periportal enrichment of E-cad was abrogated. Treatment with MSCs restored cell–cell contacts, and periportal expression of E-cad re-appeared co-localizing with β-catenin, comparable to -NASH-MSC control livers (Figure 3B). Consistent with the improvement of tissue homeostasis, MSC treatment lowered hepatic triglycerides significantly by about 40% (Figure 3C).

Taken together, MSCs thus supported restoration of tissue homeostasis by the alleviation of NASH-associated pathomechanisms like lipid load, oxidative stress, acute damage response, and EMT, corroborating results shown previously [34]. Since the WGNCA and key driver analysis suggested that the metabolic overload might be due to mitochondrial impairment, these data might corroborate the hypothesis that the MSCs might restore mitochondrial function.

### 3.3. MSCs Decreased Hepatocyte Lipid Load In Vitro

In order to study MSCs’ effects on fat-laden hepatocytes (HCs) in more detail, we established an in vitro model of hepatocyte steatosis by growing primary mouse HCs in steatosis-inducing medium, i.e., methionine-choline-deficient medium (MCD) or hepatocyte growth medium (HGM) supplemented with palmitic acid (C16:0). HGM served as a control. The mouse hepatocyte cultures in HGM and MCD medium were initially characterized in terms of the functional maintenance of hepatocyte-specific metabolism. No differences of urea synthesis were observed between the culture of cells in HGM or MCD medium. As expected [70,71], the synthesis rate decreased after one day of culture and then remained stable until 5 days of culture. The expression of selected lipid metabolism genes did not show obvious differences between culture in HGM or MCD medium. Changes over time were similar under all conditions tested. The expression of albumin was equal at each point in time, and under both culture conditions, indicating that the MCD medium did not affect hepatocyte-specific functions (Appendix A). Lipid content was quantified after 3 days of culture by image analysis of lipids after staining with Oilred O (ORO) (Figure 4A). Compared with the HGM group, MCD and C16:0 triggered a significant increase in the accumulation of lipids by 1.71- and 2.28-fold, respectively (Figure 4B). To gain further insight into the mechanisms involved in the amelioration of lipid load in livers of NASH mice by MSC treatment, HCs were co-cultured with MSCs at ratios of HCs to MSCs of 1:0, 10:1, 5:1, 1:1, or 0:1, and grown in HGM (Figure 4C), MCD (Figure 4D), or C16:0 (Figure 4E) for an additional 3 days. To note, the morphology of nuclei featured differences between HCs and MSCs, with the former displaying round-shaped and mostly binuclear with intensive DAPI staining and prominent nucleoli, and the latter presenting with oval-shaped nuclei and weaker DAPI staining. This clear distinction allowed us to readily discriminate between HCs and MSCs using fluorescence microscopy. Furthermore, the ORO staining in MSCs was undetectable under all treatment conditions (Figure 4C–E(e)), indicating that the MSCs did not accumulate lipids. When HCs were co-cultured at decreasing HC to MSC ratios, a decrease in ORO staining was observed (Figure 4C–E(a–d)). Co-culture of HCs and MSCs at a ratio of 1:1 significantly decreased the lipid load in HC from 12.56 % ± 2.7 to 6.51% ± 2.39 when cultured in HGM, from 24.46 % ± 8.75 to 7.62 % ± 4.18 in MCD medium, and from 28.67 % ± 6.66 to 17.00 % ± 2.65 in C16:0 (Figure 4F), suggesting that the MSCs supported lipid degradation in the hepatocytes. The expression of the mRNAs of Acaa1a, Acaa1b, and PPARα was higher in co-cultures in MCD medium as compared to hepatocytes alone (Appendix A), indicating that the MSCs also improved functional features in the mouse hepatocytes. This corroborated data in vivo showing a correction of lipid metabolism by MSC treatment.

MSCs may exert their effects via paracrine mechanisms involving soluble factors [72,73], or via extracellular vesicles [74]. Therefore, conditioned media (HGM, MCD, or C16:0) were collected separately from HCs, MSCs, and co-cultures of both after 48 h of culture. The media were subsequently added to HCs grown for 2 days in corresponding media and hepatocyte lipid load was determined after another 1 (Figure 4G) or 2 days (Figure 4H) by image analysis of ORO-stained lipid droplets.

A significant decrease in lipid accumulation in HCs was observed only when HCs were grown in HGM and treated with conditioned medium from co-cultures for one additional day (Figure 4G). The effect was no longer observed the day after (Figure 4H). No other condition revealed a lipid-reducing effect of the conditioned media, implicating that the lipolytic effect of MSCs in HCs was largely not mediated by soluble factors derived from MSCs, suggesting that direct cell-to-cell communication was required.

### 3.4. MSCs Communicated with Hepatocytes via Tunneling Nanotubes in a Microtubule-Dependent Manner

Irrespective of the growth in different media, MSCs communicated directly with hepatocytes in co-cultures via long filopodium-like tubes originating from the MSCs and touching the HCs or other MSCs (Figure 5A). These tubes were enriched in F-actin (cf. Figure 6A), one of the indispensable features of tunneling nanotubes (TNTs), thus potentially classifying them as TNTs. Using a live cell analysis system, we observed that the formation of TNTs was achieved by cell–cell contact between the MSCs and the targeted cell, followed by subsequent moving of the MSCs apart and leaving a tubular structure behind (Appendix A). TNTs are known to exchange molecular and corpuscular messages between cells, and mitochondria are common cargos of TNTs [75,76,77]. To understand the direction of communication between HCs and MSCs, the MSCs were pre-labeled with MitoTracker Deep Red FM and CellTrace™ Yellow prior to co-culture. The pre-labeling procedure was toxic to the HCs. Therefore, pre-labeled MSCs were co-cultured with HCs and the whole culture was stained with CellTrace™ CFSE the next day to serve as a counterstain (Figure 5B,C). The results unraveled a bi-directional cargo exchange between HCs and MSCs. The net speed of cargo transportation from HCs to MSCs (Figure 5B) and from MSCs to HCs (Figure 5C) was 627 and 1656 nm/min, respectively.

To note, though the MSCs were pre-labeled with MitoTracker to visualize the potential movement of mitochondria in TNTs, the observability was limited by the magnification and the photostability of the fluorophore. Therefore, we confirmed the involvement of human mitochondria in TNT transportation by using the anti-human mitochondria antibody, clearly indicating the transport of MSC-derived human mitochondria into the mouse hepatocytes via TNTs (Figure 6A; Appendix A). In addition, our preliminary data also show that human peroxisomes were delivered towards HCs via the TNTs (Appendix A).

The anti-AIF (anti-apoptosis-inducing factor) monoclonal antibody detects the mitochondrial antigen AIF of both mouse and human origin. AIF does not necessarily locate in the mitochondria; it is also released to the cytoplasm in response to mitochondrial membrane permeabilization [78]. In the co-culture of human MSCs and mouse hepatocytes, AIF staining was mainly enriched in HCs. Yet, also in the MSCs, AIF was detectable in patches and a weaker staining in the MSC mitochondrial network (Figure 6B). The origin and nature of the AIF in the MSCs was further confirmed by using the anti-human mitochondria antibody and MitoTracker. The patchy AIF in the MSCs co-localized with MitoTracker but was negative for anti-human mitochondria staining (Figure 6B(a,b)), suggesting that these structures were derivatives of mitochondria of mouse origin. At present, we cannot say that mouse mitochondria in the MSCs were delivered via the TNTs, but the results corroborate the assumption of a bi-directional exchange between HCs and MSCs. The functional meaning, however, remains elusive and needs further investigations.

To further confirm the character of the TNTs and unravel potential functional consequences, we examined the expression of relevant molecular motors or proteins by RT-PCR using human-specific primers in association with the two most studied cytoskeleton proteins related to TNT delivery, tubulin (representing microtubule-based transport) and actin (representing actin-based transport). Compared with MSC mono-cultures, the mRNA expression of human microtubule-associated proteins, namely Ras homolog family member T1 (Rhot1, also known as mitochondrial Rho GTPase 1; MIRO1) and kinesin family member 5B (KIF5B), were upregulated in co-cultures (Figure 7A). However, neither inducers of actin-based TNTs, RAS like proto-oncogene A (RALA) and TNFα-induced protein 2 (TNFAIP2), were altered. The results implied that in the co-culture with mouse hepatocytes, human MSCs may facilitate the expression of the microtubule-related but not actin-related gene expression to foster human MSC-derived TNT-dependent cargo transport.

### 3.5. MSCs May Have Enhanced Lipid Utilization in HCs by Eliciting Oxidative Capacity

We hypothesized that the transportation of organelles from MSCs to HCs may promote lipid utilization in HCs in two ways: 1) By the activation of the key regulators of lipid utilization, like, e.g., peroxisome proliferator-activated receptor α (PPARA), and/or 2) by increasing the lipid-oxidizing capacity in HCs by the increase in the amount of MSC-derived mitochondria. To discriminate between human and mouse effects, we used species-specific primer pairs in RT-PCR experiments. The expression of mouse PPARA, but not human PPARA, was significantly higher in co-culture than in HC mono-culture (Figure 7B), corroborating the gene array results as shown above (Appendix A) and further suggesting that MSCs may elicit the utilization of lipids and fatty acids in HCs. Next, we analyzed the expression of markers involved in mitochondria biogenesis like PGC1α, mitochondrial transcription factor A (TFAM), and heme oxygenase-1 (HMOX1) by species-specific RT-PCR to discriminate between mouse and human effects. Only the expression of mouse PGC1α (PPARGC1A; PPARγ coactivator 1 α), a regulator of mitochondria biogenesis, was significantly enhanced in co-cultures with MSCs as compared with HCs alone (Figure 7C). Albeit not significantly, there was a trend of higher expression of mTFAM, while mHMOX1 was even decreased if affected at all. No human markers of mitochondria biogenesis were changed in co- vs. mono-cultures (Figure 7C). Therefore, also taking the results as shown in Figure 8A,B into account, it may be suggested that the MSCs might promote mitochondria biogenesis in the mouse hepatocytes. This, however, needs further confirmation.

Neither the number (Figure 8A) nor the area (Figure 8B) of mitochondria in MSCs and HCs were altered by any treatment. Yet, albeit not significant, there was a trend of higher mitochondria numbers in HCs when co-cultured with MSCs, both in HGM and in MCD medium, suggesting that MSCs might increase the number of mitochondria in HCs, in line with data shown in Figure 7C. The percentage of one of the mitochondrial subtypes, small globules, was significantly decreased in HCs when cultured in MCD medium, which was reversed in part in co-cultures with MSCs (Figure 8C). When cultured in MCD medium, the percentage of small globules was significantly increased in MSCs, suggesting that this condition might increase specific subtypes of mitochondria in MSCs (Figure 8D). Taking into consideration the mitochondrial globular shape as shown in Figure 6A in co-cultures of HCs and MSCs, it was very obvious that preferentially small dotted (small globules) MSC-derived mitochondria were detectable in TNT bridging to HCs (Figure 8E).

Taken together, these results show that MSCs may deliver mitochondria (and bona fide peroxisomes) to HCs, which might support lipid breakdown both by providing oxidative capacity and by support of the hepatocytes’ own capacity of lipid utilization potentially by the support of mitochondria biogenesis. However, the mutual interactions between recipient and donor mitochondria and the impact on the recipient lipid metabolism remains to be further elucidated.

### 3.6. Human BM-MSCs Delivered Mitochondria to Mouse Hepatocytes In Vivo

In order to show that human mitochondria from transplanted MSCs were delivered to mouse hepatocytes after hepatic transplantation in vivo, liver slices were co-stained with an anti-mouse-specific cyclophilin and the anti-human-specific mitochondria antibody. Image analysis using the Zeiss Axio Observer.Z1 microscope equipped with ApoTome.2 showed that human mitochondria were only detectable in livers, which received human MSC transplants (Figure 9A). Signals co-localized with cyclophilin (Figure 9B), clearly indicating that donor human MSC-derived mitochondria were delivered to the host mouse hepatocytes. At this point in time, these results would be in line with the hypothesis that the delivery could involve TNTs as demonstrated in vitro (cf. Figure 6A).

## 4. Discussion

### 4.1. Pathobiochemical Consequences of Changes in Metabolic Protein Expression and Correction by MSCs

Based on the proteomics and metabolomics data, the WGCNA predicted a MCD diet-induced deregulation of central carbon and amino acid metabolism likely associated with mitochondrial dysfunction, which were reversed by MSC treatment, rendering most of the affected metabolic pathways not significantly different from controls.

NASH is characterized by the accumulation of triglycerides in hepatocytes, a predominant sign of metabolic impairment. In the MCD model of NASH, it has been shown that VLDL secretion is inhibited due to the attenuated expression of the chaperone microsomal triglyceride transfer protein (MTP), which is essential for the proper folding of apolipoprotein B (ApoB) [79]. Here, we showed that ApoB was slightly increased, likely due to the accumulation of the non-functional protein, while the expression of ApoA4, involved in VLDL secretion [80], was downregulated. The increased expression of hepatocyte fatty acid transporters Fatp1 and CD36 fostered fatty acid uptake and together with the perturbance of lipid (lipoprotein) secretion likely caused an imbalance of lipid metabolism favoring storage over secretion. In addition to the attenuation of triglyceride secretion, the utilization of fatty acids by mitochondrial β-oxidation seemed to be impaired by the downregulation of proteins of the respiratory chain as obvious from the key driver analysis. Further, at the individual level of expressed genes, Acaa1a and b, involved in peroxisomal fatty acid oxidation, were downregulated, indicating an additional impairment of peroxisomal lipid utilization. In total, these findings are consistent with NASH-induced changes in central carbon and amino acid metabolism (as side reactions associated with glycolysis and the tricarboxylic acid cycle [81]) promoting lipid storage due to failure of lipid utilization and secretion as commonly observed in rodent NASH models and in humans [82,83,84].

Taken together, the pathobiochemical changes in the mouse liver upon MCD diet feeding are in line with major features of NASH as described [85,86]. In the study presented here, MSCs ameliorated the hepatic lipid load consistent with the improvement of NASH-induced metabolic changes as unraveled by WGCNA. To our knowledge, this is the first study to show the impact of MSCs on key metabolic pathways involved in the pathogenesis of NASH. However, the molecular and/or cellular mechanism(s) engaged remain open. Since it is conceivable that the MSCs did not impact on each individual biochemical pathway separately, or even by different mechanisms, it may be assumed that the MSCs primarily attenuated the hepatocyte lipid load by a unique mechanism, and consecutively improved overall metabolic homeostasis.

### 4.2. Histopathological Consequences of Hepatocyte Lipid Load and Improvement by MSCs

The accumulation of lipids in hepatocytes is associated with an overproduction of reactive oxygen species (ROS) that progressively exceeds the cellular detoxification capabilities. Microsomal Cyp2e1 is a major site of fatty acid metabolism, which has been found to be elevated in fatty liver diseases in humans and rodents [66]. Cyp2e1 plays a critical part in the pathogenesis of NASH by the production of ROS, which foster protein and lipid peroxidation associated with cellular stress and damage [67]. This is in line with our findings of increased Cyp2e1 and elevated 4-HNE as a byproduct of lipid peroxidation in the NASH livers. As identified by the key driver analysis in the NASH livers, the upregulation of proteins involved in protection from oxidative stress like Mgst1 and Gstp1 [87] may be interpreted as an adaptive response to increased oxidative stress, while downregulation of proteins like Sod2 involved in mitochondrial superoxide detoxification may even aggravate oxidative stress and mitochondrial impairment [88].

Alpha-1-antitrypsin and major urinary proteins increased in the NASH livers, indicative for the onset of the acute-phase reaction, the hepatic defense response to injury and inflammation. Likewise, the expression and localization of the cell adhesion proteins E-cadherin and β-catenin changed in the NASH as compared to control livers, indicative for the perturbation of the epithelial organization of the hepatic parenchyma. Epithelial-mesenchymal transitions in the liver have been attributed to be essential in regeneration of the liver after injury [89]. Hence, tissue deterioration in the NASH livers may indicate regeneration due to hepatocellular death.

MSC ameliorated oxidative stress, inflammation, and tissue damage, corroborating our previous results showing regression of fibrosis and cell death as well as attenuation of inflammatory pathways [20], consistent with findings demonstrating resolution of inflammation, fibrosis, and lipid load in rodent models of NASH [90,91]. The mechanisms behind, however, remain elusive. Again, it is unlikely that the MSCs impacted on single individual pathways involved in the perturbation of tissue homeostasis by different modes of action. Therefore, we hypothesize that the overall improvement of tissue homeostasis secondarily followed the attenuation of the hepatocyte lipid load. This implies that metabolic improvement preceded the restoration of tissue architecture, which was likely to be achieved by the regenerative capacity of the liver itself.

### 4.3. Mitochondrial Transfer by MSCs May Attenuate Hepatocyte Lipid Load

Previous reports including mouse datasets into WGCNA approaches identified mitochondrial dysfunction as a key event in the pathogenesis of NASH [92], consistent with our data that aberrant expression of mitochondrial proteins may drive mitochondrial dysfunction. Indeed, in consequence of insulin resistance, the increased flux of free fatty acids from the adipose tissue to the liver has long been suggested as a major cause of the metabolic overload and eventually dysfunction of mitochondria in the pathogenesis of NASH [7,8,93]. Based on these facts, we hypothesized that the improvement of mitochondrial function in the host liver by transplanted MSCs may play a central role in the amelioration of lipid load in the NASH livers. In the in vitro experiments, we observed a physical long-distance connection between MSCs and hepatocytes. These protrusions contained filamentous actin and carried whole organelles, features that have been specifically attributed to phenomena called tunneling nanotubes (TNTs) [94,95]. TNTs are thin membranous structures mediating direct intercellular communication between non-adjacent cells. They have been shown to transport subcellular components, e.g., mitochondria and lysosomes as well as plasma membrane components, mRNA, electrical signals, and calcium ions [96]. Our in vitro data present evidence for a transfer of mitochondria from MSCs to the hepatocytes via TNTs. Since the steatosis-inducing media used here is known to cause mitochondrial dysfunction associated with excess lipid storage, ROS generation, and cytotoxicity [97,98], the donation of functional mitochondria by the MSCs might contribute to the attenuation of lipid load in the hepatocytes. Hence, we suggest that the transferred mitochondria may compensate for a lack of oxidative phosphorylation capacity and thereby improve lipid utilization, which is in line with the lipid-lowering effect of MSCs as shown both in vitro and in vivo. Our work does not provide direct evidence that the delivery of mitochondria is involved in the reduction of triglycerides in hepatocytes. We used inhibitors of the respiratory chain and microtubule assembly in order to demonstrate correction of hepatocyte metabolism by MSC-derived donor mitochondria. Yet, most of the inhibitors were toxic to the hepatocytes, thus not allowing to discriminate between toxicity and specific effects on targets to be inhibited. It may not be excluded that part of the hepatocyte lipid resources may be metabolized by the MSC transplants. This would require the transfer of lipids to the MSCs, which we neither observed in vitro nor in vivo. Additionally, MSCs may have stimulated lipid breakdown in the host hepatocytes to release free fatty acids, which then could be metabolized by the MSCs. This would require some paracrine mechanisms, which, however, we may exclude, because MSC-derived conditioned medium was not effective. In addition, MSCs did not accumulate lipids in vitro, indicating that their lipid metabolism might be different from the hepatocytes. Indeed, in their niche, MSC energy metabolism may primarily rely on anaerobic glycolysis rather than oxidative phosphorylation [99]. Our results are consistent with data shown in a high-fat diet-fed mouse model, in which MSC transplants improved mitochondrial morphology and metabolic performance in the host liver [100]. This may describe a general mechanism of MSC action, since stem cell-derived donor mitochondria increased oxidative phosphorylation and ATP generation, and reduced ROS production in host cells like, e.g., myeloma cells [101], macrophages [102], cardiomyocytes [103], and airway epithelial cells [104]. Thus, the concept of MSC-derived mitochondrial transfer to improve recipient cell oxidative metabolism is now well established [105]. However, in our model of NASH, mechanistic questions remain open: How are diseased mitochondria cleared from the hepatocytes (e.g., by mitophagy)? Which are the mechanisms protecting the donated healthy mitochondria? How is the pool of donor mitochondria replenished in donating MSC? These questions are topics of our current work.

In our mouse model, we saw that MSC transplants entered the liver parenchyma primarily at the periportal area of the liver sinusoid, resulting in zonal enrichment of the MSCs [20]. For this animal model, we calculated that a 1% repopulation of the hepatic parenchyma by transplanted MSCs may be achieved, i.e., 6.6 × 10^5^ transplanted cells in the host liver [106]. Assuming that a cell on average may contain 1500 mitochondria [107], this cell number corresponds to 9.9 × 10^8^ donor mitochondria. A mouse liver contains 66 × 10^6^ hepatocytes [108] corresponding to 9.9 × 10^10^ mitochondria. In conclusion, 1% on average of all mitochondria in a mouse hepatocyte ought to be of human MSC donor origin. This is in the same order of magnitude as shown in a co-culture model of human bone marrow-derived MSCs and myeloma cells, which improved cellular respiration in the range of 10–50% depending on the cell line under investigation [101]. If we assume in a simple approximation that this increase would solely contribute to lipid degradation and oxidation, this would account for the decrease in hepatocyte lipid content as shown here in vivo and in vitro of roughly 40%.

## 5. Conclusions

Metabolically, the diet-induced NASH was characterized by an impairment of the central carbon metabolism likely due to mitochondrial and peroxisomal dysfunction. The enhancement of membrane transporters of fatty acids may have increased fatty acid uptake. In association with the impairment of utilization and lipoprotein secretion, the enhancement of triglyceride synthesis and storage may follow. MSCs, transplanted into a mouse host liver, ameliorated lipid storage and associated perturbance of tissue homeostasis likely by the donation of healthy mitochondria to the hepatocytes via TNTs, thus providing oxidative capacity for lipid breakdown and eventually restoration of tissue homeostasis.

## Figures and Tables

**Figure 1 biomedicines-08-00350-f001:**
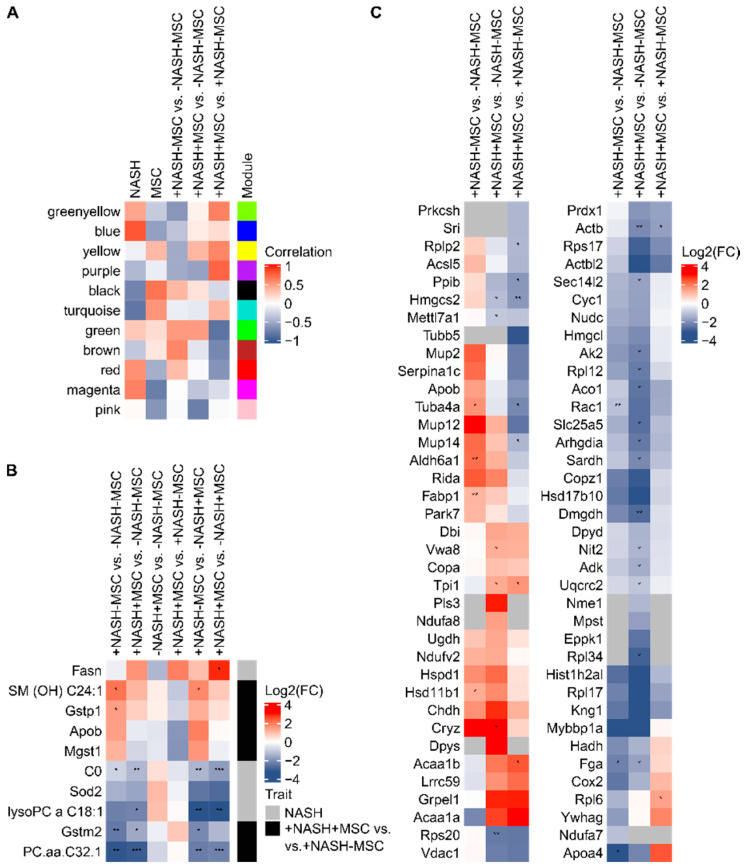
Integrative analysis of proteome and metabolome. (**A**) Correlations of WGCNA-created modules with relevant traits. Based on this, key drivers for the traits NASH and +NASH+MSC vs. +NASH-MSC were identified; (**B**) Log2(FCs) and *p*-values for selected key drivers; (**C**) Log2(FCs) and *p*-values are presented for selected candidates, which showed differences, albeit not significant throughout, between +NASH-MSC and +NASH+MSC. Significances are indicated with asterisks (*p*-value ≤ 0.05: *, *p*-value ≤ 0.01: **, *p*-value ≤ 0.001: ***). Euclidean clustering was applied to all heatmaps shown.

**Figure 2 biomedicines-08-00350-f002:**
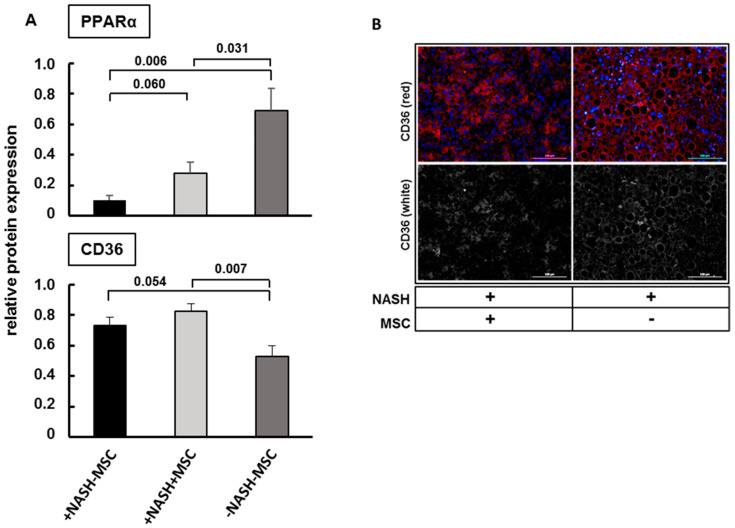
Expression of CD36 and PPARα in MCD-treated mice with and without MSC application. (**A**) Upregulation of CD36 expression and downregulation of PPARα in NASH livers and partial reversal by MSCs as shown by semiquantitative Western blot analysis of liver cytosolic extracts from 6 different animals in each group. Vinculin was used for normalization. Values represent means ± SEM and significant differences as indicated by the *p*-values over the horizontal lines were identified by applying the Student’s *t*-test for unpaired values; (**B**) fluorescent immunohistochemical detection of CD36 in representative liver slices of +NASH-MSC livers and after treatment with MSC (+NASH+MSC). Scale bar 100 µm.

**Figure 3 biomedicines-08-00350-f003:**
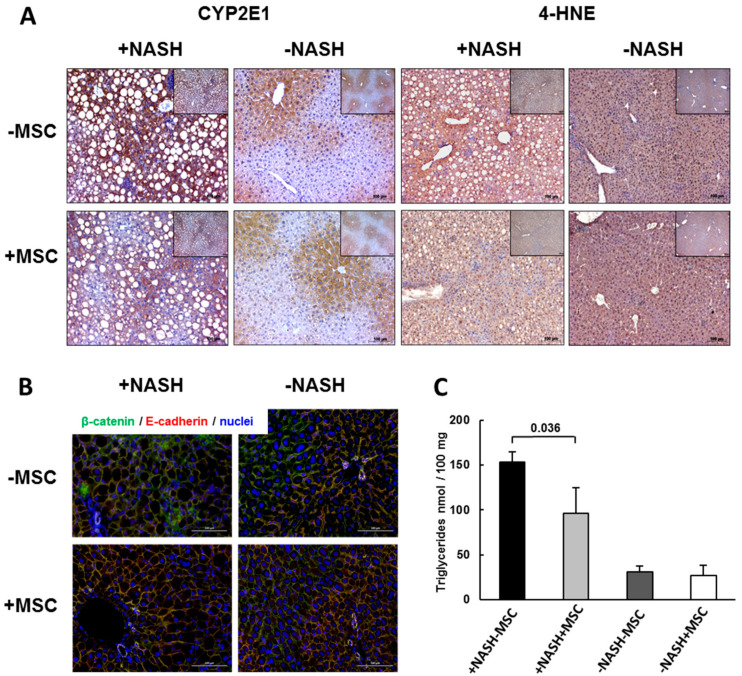
Liver tissue deterioration in MCD-treated mice and reversal by MSCs. (**A**) Immunohistochemical detection of Cyp2e1 and 4-HNE reveals an increase in perivenous localization in NASH livers, and restoration of zonation to nearly normal by MSC treatment. Pictures are representative for 3 different animals out of each group. White “holes” represent lipid droplets in NASH livers (original magnification 10×). The insets show lower magnifications for an overview impression (original magnification 5×). (**B**) Fluorescent immunohistochemical detection of adherens junction proteins β-catenin (green fluorescence) and E-cadherin (red, yellow in the overlay) indicates periportal zonation of E-cadherin, which is lost in NASH livers and restored by treatment with MSCs. Pictures are representative for 3 different animals out of each group. Scale bar 100 µm. (**C**) Hepatic triglycerides in control (-NASH) and in MCD diet-fed (+NASH) mice either treated without (-MSC) or with (+MSC) human bone marrow-derived MSCs. Values are means ± SD from 5 animals in each group. The horizontal line indicates the significant difference between the +NASH-MSC and the +NASH+MSC group at *p* = 0.036. In addition to Johnson transformation, an ANOVA was performed *p* = 0.0001, post-hoc Dunett T *p* = 0.024.

**Figure 4 biomedicines-08-00350-f004:**
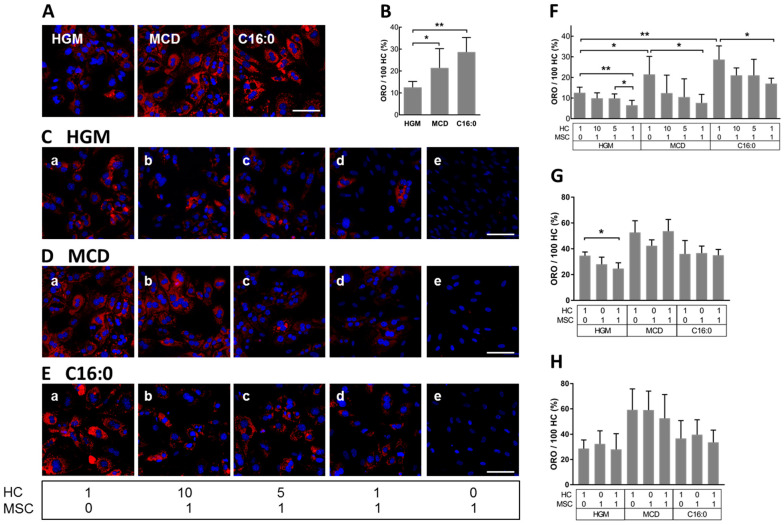
Induction of lipid droplet formation in cultured hepatocytes (HCs) by steatosis-inducing medium and reversal by MSCs in co-culture. Primary HCs were either cultured in HGM medium or treated with steatosis-inducing MCD medium or HGM supplemented with 0.5 mM palmitic acid (C16:0) for 3 days. (**A**) Visualization of hepatocyte lipids with Oil red O (red) and (**B**) quantification. HCs cultured (**a**) alone or together with MSCs at ratios of HCs to MSCs of (**b**) 10:1, (**c**) 5:1, (**d**) 1:1, or (**e**) 0:1 were grown for 3 days either in (**C**) HGM, or in (**D**) MCD, or in (**E**) HGM supplemented with C16:0. (**C–E**) lipid stain and (**F**) quantification. Nuclei were counterstained with DAPI (blue). Conditioned media were collected from either HC (1:0) or MSC (0:1) mono-cultures, or co-culture (ratio 1:1) and transferred to HCs grown for an additional (**G**) 1 or (**H**) 2 days in either HGM, MCD medium, or HGM supplemented with C16:0. The lipid stain with Oil red O was quantified by image analysis and the results from 3 independent cell cultures were normalized as the percentage amount of stain/100 hepatocytes and expressed as mean ± SD. Statistical comparisons were made using unpaired *t*-tests, and differences between groups were considered significant if the *p*-value was ≤0.05. *: *p* ≤ 0.05; **: *p* ≤ 0.01. Scale bar 100 μm. MSCs: human bone marrow-derived mesenchymal stromal cells; HCs: mouse primary hepatocytes; HGM: hepatocyte growth medium; MCD: methionine-choline-deficient medium.

**Figure 5 biomedicines-08-00350-f005:**
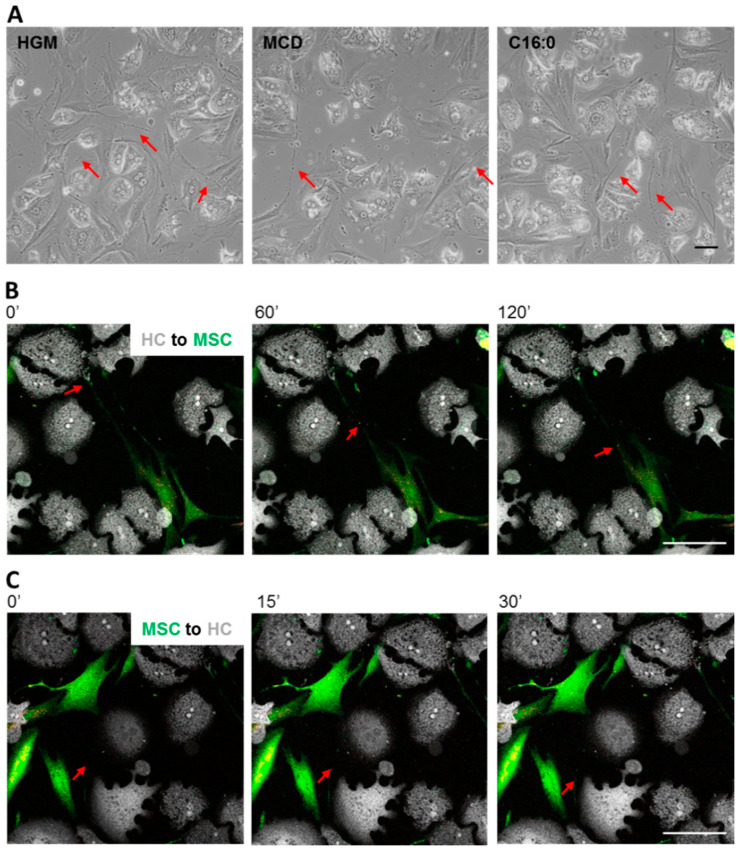
TNT-mediated cargo exchange between HCs and MSCs. (**A**) On day 1 of co-culture, the TNT structures (red arrows) derived from MSCs were readily detectable by phase contrast microscopy in cultures grown under all tested conditions. Scale bar 100 µm. Corresponding movies may be opened in Supplementary Material file 3; (**B**) The delivery of cargos from HCs to MSCs and (**C**) from MSCs to HCs was monitored by co-culture of HCs and MSCs pre-labeled with CellTrace™ Yellow and MitoTracker™ Deep Red FM (pseudo-colored green and red, respectively). The whole culture was stained with CellTrace™ CFSE (pseudo-colored white) and the pictures were captured using time-lapse confocal imaging. When the first picture was taken, this time point was designated as time 0, to which the other time points refer. The direction of movement of cargos in the TNTs is indicated by the red arrows. Scale bar 100 µm. Higher magnification images are available in Appendix A. Corresponding movies may be opened in Supplementary Material file 4.

**Figure 6 biomedicines-08-00350-f006:**
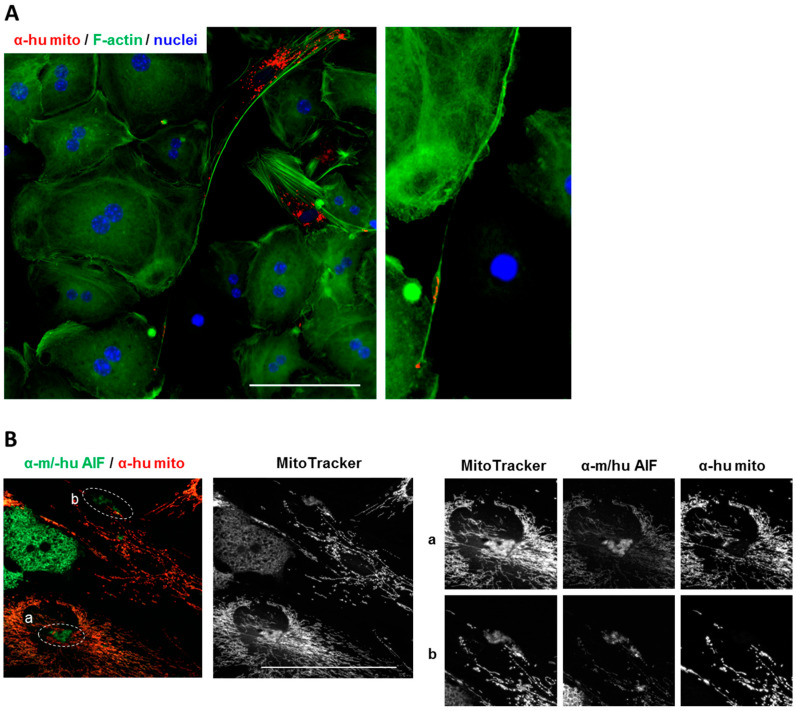
TNTs between HCs and MSCs are used to transport mitochondria. (**A**) Human MSC-derived mitochondria, stained in red with the anti-human-specific antibody against human mitochondria, are delivered to co-cultured mouse hepatocytes (mostly bi-nucleated). F-actin was stained with Phalloidin-iFluor 488 (green), nuclei with DAPI. Scale bar; 100 µm. Right panel: Computational enlargement of an area as shown on the left panel; (**B**) Mouse and human mitochondrial apoptosis-inducing factor (AIF) (green) and human mitochondria (red) were detected by fluorescent immunocytochemistry using species-specific antibodies, and cells were further stained with MitoTracker™ Deep Red FM (white). (**a**) and (**b**) show higher magnification pictures (computational enlargements) of circled areas shown in the panels on the left. Scale bar 100 μm.

**Figure 7 biomedicines-08-00350-f007:**
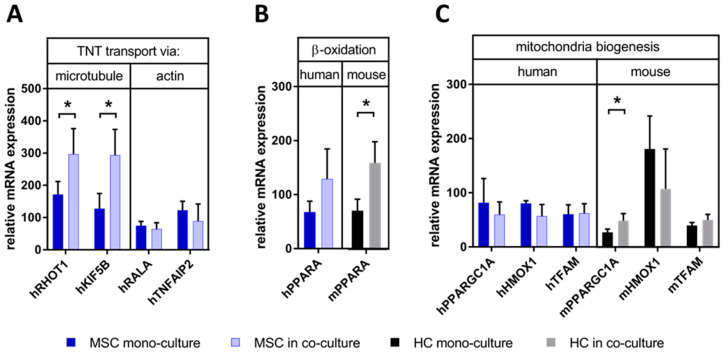
Expression of factors involved in (**A**) microtubule- and actin-based tubular transport, (**B**) hepatocyte lipid utilization, and (**C**) mitochondria biogenesis. Expression levels were analyzed by RT-PCR using species-specific primer pairs (blue/light blue columns for the use of human (h) primers, black/grey columns for the use of mouse (m) primers) and mRNA levels normalized with beta-2-microglobulin. Results are expressed as mean ± SD. Statistical comparisons from 3-5 independent cell cultures were made using the 2-way ANOVA test after log transformation, and differences between groups were considered significant if the *p* value was ≤0.05 (*). h: human; m: mouse; RHOT: Ras Homolog Family Member T1, also known as mitochondrial Rho GTPase 1 (MIRO1); KIF5B: kinesin family member 5B; RALA: RAS like proto-oncogene A; TNFAIP2: TNFα-induced protein 2; PPARGC1A: PPARα coactivator 1α, also known as PGC1α; HMOX1: heme oxygenase-1; TFAM: mitochondrial transcription factor A; PPARA: peroxisome proliferator-activated receptor α.

**Figure 8 biomedicines-08-00350-f008:**
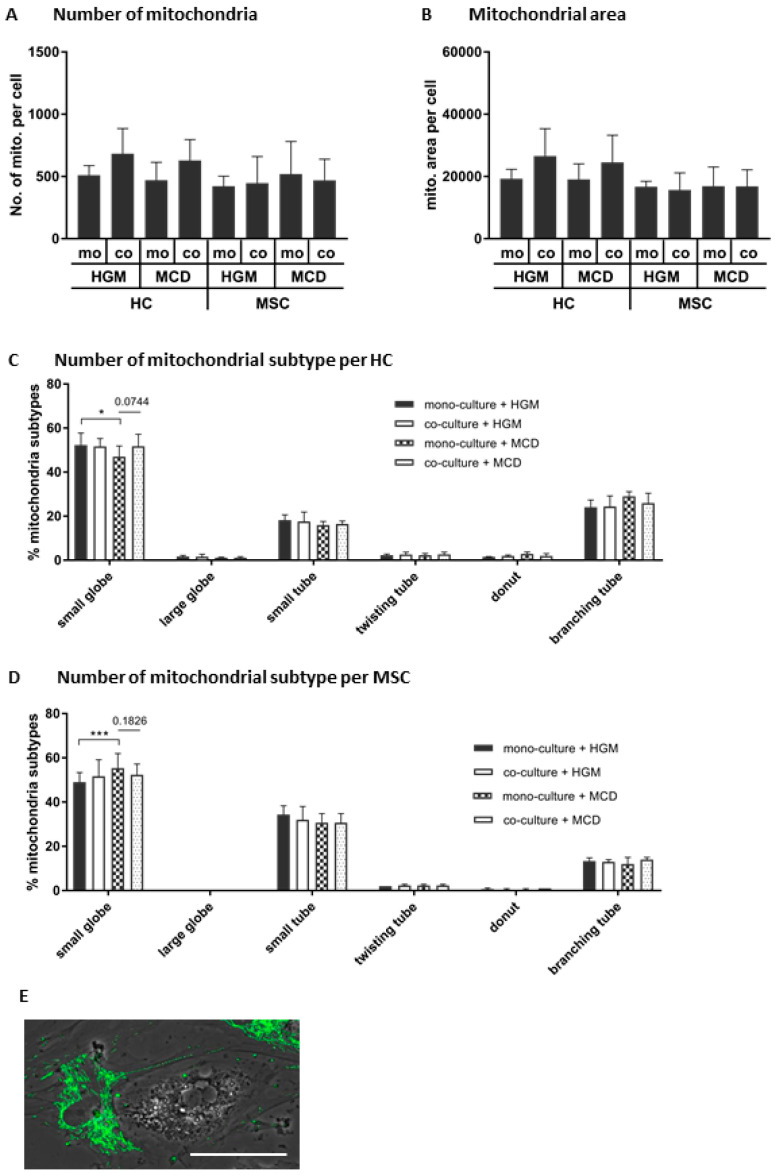
Profiling of cell-type-specific mitochondria in co-cultures of mouse hepatocytes and human MSCs. On day 1 of co-culture, fluorescence images of cells stained with MitoTracker Red CMXRos were analyzed for the (**A**) number and (**B**) area of mitochondria and percentages of mitochondria subtypes (**C**) in HCs and (**D**) in MSCs using the software MicroP. The statistical comparisons from 4 independent cell cultures were made using the 2-way ANOVA test. Results are expressed as mean ± SD. Statistical comparisons were made using unpaired *t*-tests, and differences between groups were considered significant for the *p*-values *: *p* ≤ 0.05; ***: *p* ≤ 0.001. (**E**) The globular morphology of MSC mitochondria was confirmed (cf. also Appendix A) in MSCs co-cultured with HCs in HGM or MCD medium. The mitochondria were stained (pseudo-colored in green) and the picture was merged with the light microscopy image. Scale bar 50 µm. mo: mono-culture; co: co-culture.

**Figure 9 biomedicines-08-00350-f009:**
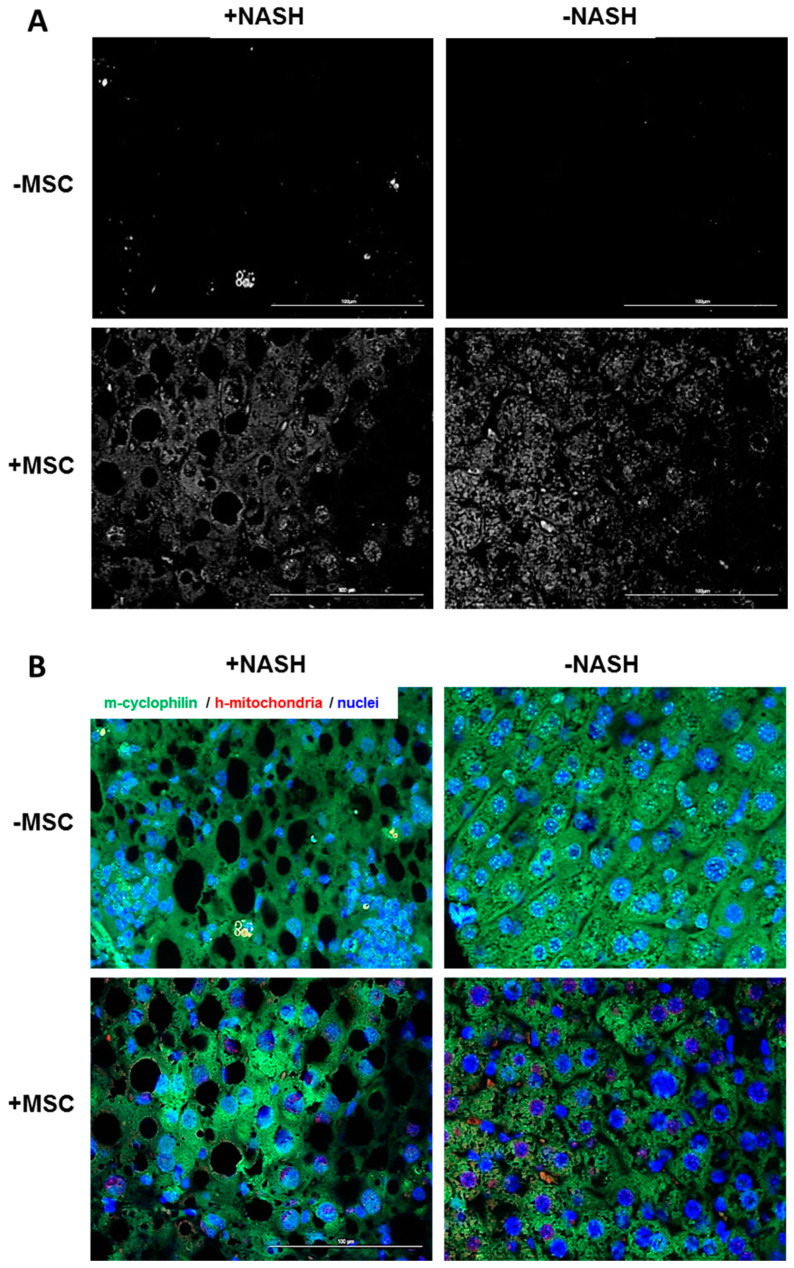
Human MSC-derived mitochondria in mouse hepatocytes of animals receiving MSC transplants. Here, 2-µm slices of mouse livers either fed the control (-NASH) or the MCD diet (+NASH) and treated without (-MSC) or with (+MSC) human bone marrow-derived MSC were co-stained with the anti-mouse cyclophilin or the anti-human mitochondria antibody and images captured using the Zeiss Axio Observer.Z1 microscope equipped with ApoTome.2 with a 40× objective. (**A**) Black and white images of pictures shown in (B) indicate human mitochondria (lower panels) in livers, which were transplanted with human MSCs. Non-transplanted livers (upper panels) were void of signals; (**B**) Immuno-fluorescent co-stain of mouse cyclophilin (green channel), and human mitochondria (red channel) indicating human mitochondria in mouse hepatocytes; nuclei were stained with DAPI (blue).

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
