# Peer review of "Mitochondrial Transfer by Human Mesenchymal Stromal Cells Ameliorates Hepatocyte Lipid Load in a Mouse Model of NASH"

_biomedicines, 2020, doi:10.3390/biomedicines8090350_

Round 1

Reviewer 1 Report

The manuscript of Hsu et al. describes the identification of a mitochondrial dysfunction in both a mouse model of hepatic steatosis and a co-cultured cell model.  The authors then provide intriguing novel evidence for a mechanism of mitochondrial transfer from donor MSC cells to mouse hepatic cells via tunnelling nanotubes that can promote triglyceride degradation and regression of the steatosis.  Overall, this is an intriguing manuscript with sufficient preliminary evidence to support their conclusions.  I have a number of comments listed below.   1. page 2 line 40: There is very little evidence to support transmission of peroxisomes in the data presented.  This will have to be removed from most places.   2. Section 3.1 The authors have presented a very technical description of the omics approaches.  The authors may be able to provide a better understanding of this section to the general audience if they made a few general "lay statements" about procedures and results.     3. At the beginning of Section 3.1, it is unclear what the authors are testing.  Are they analyzing microarray or proteomic data here?  I am assuming the authors are using proteomic data on the mouse livers described in the methods (and should state here explicitly) but then they use some gene names instead of protein names.    4. page 8, line 13-14: The authors state that the key drivers of lipid metabolism appear to be apoB and FASN.  There are many proteins that are involved in lipid metabolism, and so it would be helpful if the authors could dissect the data of some of these key drivers to show a general trend for all of the proteins involved. Otherwise it would just be "cherry-picking".   For example, if we wanted to comment on impaired VLDL secretion in one of the models (p.9 line 17-19), we would analyze apoB, apoA4, MTTP, CIDEB, SOAT2, DGAT2, as well as many potential lipid droplet associated proteins.  Rather than just commenting on apoB or apoA4 and saying VLDL secretion is likely impaired, I would appreciate a more thorough approach, especially considering that you have all the proteomic data already.  This should be done for acute phase reaction (p.9 line 1), mitochondrial proteins (p.9, line 8), peroxisomal proteins, triglyceride synthesis (p.9, line 12), and VLDL secretion (p.9, line 17-19).   Otherwise the conclusion (p.9, line 19-22) is not justified.  I was intrigued that some COP components (COPA and COPZ1) were affected (were any others?) as there is some evidence in the literature that these affect lipid droplet dynamics and VLDL lipidation.   5. Figure 2A: the western blot does not demonstrate any significant difference in PPARa between the two conditions.  This is certainly not strong enough evidence to conclude that peroxisomal activity could contribute to lipid clearance (p.9, line 33).   6. Page 10, line 7-9: To be fair, in order to make this claim, the authors would need to demonstrate biochemically that fatty acids degradation is being stimulated.  Was lipophagy stimulated?  Was TG secretion within VLDL affected?     7. p.5 line 47: "aq. dest."  Does this mean distilled water?  This is not a widely used convention and should be changed to just "distilled water".   8. metabolization should be metabolism.   9. Figure 3C, Y-axis figure legend is misspelled.

Reviewer 2 Report

The manuscript by Hsu et al. entitled “Mitochondrial Transfer by Human Mesenchymal Stromal Cells Ameliorates Hepatocyte Lipid Load in a Mouse Model of NASH” represents data from an interesting and technically sophisticated study. The authors have been focused on addressing the question regarding the mechanisms that underlie an improvement of metabolic phenotype in diet-induced non-alcoholic steatohepatitis (NASH) in mice that undergone transplantation of mesenchymal stromal cells (MSC). By using a number of state-of-the-art molecular analysis approaches and visualizing techniques, the authors generated data allowing to conclude that MSC, transplanted into a mouse host liver, can ameliorate lipid storage and associated perturbance of tissue homeostasis by the donation of mitochondria to the hepatocytes, thus providing oxidative capacity for lipid breakdown and eventually restoration of tissue homeostasis.  Undoubtedly this is a well written and rigorously presented research manuscript with data that contributes significantly to the field of metabolic disease and will be of interest to the research community.

At the same time there are several methodological and conceptual concerns requiring clarification:

  1. The major concept of the study is the transfer of mitochondria from MSC to hepatocytes that results in increased oxidative capacity and thus improvement of the steatotic phenotype. Although it is not clear whether MSC themselves (without transferring their mitochondria) may contribute to the increased lipid oxidation in the livers or in co-cultures by metabolizing fatty acids, thus diminishing hepatocyte lipid overload. In this context experiments employing knockdown or silencing tunneling nanotubes (TNT) or microtubule component(s) would be highly important to address this question and resolve the possible controversy.
  2. Because NASH in vivo is not always an isolated local (hepatic) event and rather is associated with broader systemic metabolic alterations (dyslipidemia, glucose intolerance, weight gain), it is interesting and worth presenting and discussing whether and how other parameters associated with metabolic disease, including plasma lipidemia, glucose clearance, and adiposity, are affected in the current model.
  3. Immunoblots representing CD36 protein expression (Fig. 2A) do not look convincing because of the weakness (this also relates to PPARα) and high variability of the signal. Moreover, which of the two bands around 75kDa is considered to be CD36? These blots should be improved.
  4. The images of liver sections stained for CYP2E1 and for 4-HNE (Fig. 3A) seemed to be taken at different magnification because they are clearly different by the degree of steatosis (as evidenced by the white “holes”). Moreover, the section stained for 4-HNE in the +NASH+MSC group looks more steatotic than in the +NASH-MSC group, which contradicts the whole concept of the study.
  5. In culture experiments (Fig. 4), hepatocytes are cultured for about 6 days, and thus may be associated with hepatocyte dedifferentiation. To resolve this concern the authors should include data on certain marker hepatocyte parameters like albumin expression/secretion, urea production, or CYP expression.
  6. It is an interesting point to suggest that the MSC may promote mitochondria biogenesis in the mouse hepatocytes (p.16-17). However, how accurate is it to suggest this based on the changes in Pgc1a mRNA levels, especially when mRNA expression of other studied mitochondria biogenesis genes remains unchanged. What about protein levels? What about Prdm16 and Ucp1 expression?
  7. Line 10 on page 17 implies that peroxisomes are being transferred to hepatocytes from MSC as well, however, which data specifically support that?

Round 2

Reviewer 2 Report

All of the concerns have been fully addressed.